# The Influence of Left-Behind Experience on College Students’ Mental Health: A Cross-Sectional Comparative Study

**DOI:** 10.3390/ijerph17051511

**Published:** 2020-02-26

**Authors:** Haixia Liu, Zhongliang Zhou, Xiaojing Fan, Jiu Wang, Hongwei Sun, Chi Shen, Xiangming Zhai

**Affiliations:** 1School of Public Health, Xi’an Jiaotong University Health Science Center, No. 76 Yanta West Road, Xi’an 710061, China; liuhaixia127@stu.xjtu.edu.cn; 2School of Public Health and Management, Binzhou Medical University, No. 346, Guanhai Road, Laishan District, Yantai 264003, China; mmswj@163.com (J.W.); hwsun2000@163.com (H.S.); 3School of Public Policy and Administration, Xi’an Jiaotong University, No. 28 Xianning West Road, Xi’an 710049, China; fanxj112@xjtu.edu.cn (X.F.); shenchi@stu.xjtu.edu.cn (C.S.); xiangmingzhai@stu.xjtu.edu.cn (X.Z.)

**Keywords:** left-behind experience, social support, college students with left-behind experience, mental health problems, coarsened exact matching

## Abstract

China’s rapid development and urbanization have created large numbers of migrant laborers, with increasing numbers of young adults and couples migrating from rural areas to large cities. As a result, a large number of children have become left-behind children (LBC), who were left behind in their hometown and cared for by one parent, grandparents, relatives or friends. Some of these LBC have a chance to be college students, who are called college students with left-behind experience. Some studies have indicated that the absence of these college students’ parents during childhood may cause them to have some mental health problems. Therefore, we want to examine the effects of left-behind experience on college students’ mental health and compare the prevalence of mental health problems in left-behind students and control students (without left-behind experience). For this purpose, a cross-sectional comparative survey was conducted in a coastal city of Shandong province, Eastern China. First, 1605 college students from three universities (national admissions) were recruited, including 312 students with left-behind experience and 1293 controls. Their mental health level was measured using Symptom Check-list 90 (containing ten dimensions: somatization, obsessive-compulsion (OCD), interpersonal sensitivity, depression, anxiety, hostility, terror, paranoia, psychoticism, and other symptoms). The results showed that left-behind experience was a significant risk factor for the mental health problems of college students (*OR* = 2.27, 95%*CI*: 1.73 to 2.97). A comparison of the two groups, after controlling the confounding factors using the coarsened exact matching (CEM) algorithm, showed that the prevalence of mental health problems was 35.69% (n = 311) among the left-behind students, while it was 19.68% (n = 1194) among the controls. The two groups were significantly different in terms of these ten dimensions of the SCL-90 scale (*p* < 0.001), and the prevalence of each dimension among the left-behind students was consistently higher than that among the controls. In addition, different left-behind experiences and social supports during childhood had different effects on mental health problems.

## 1. Introduction

The past four decades of modernization and urbanization in China have created large numbers of migrant laborers, with increasing numbers of young adults and couples migrating from rural areas to large cities. These adults usually do not take their children with them due to the high living costs and barriers to accessing educational services in the cities where they work [1]. As a result, a large number of children become left-behind children (LBC), who are left behind in their hometown, cared for by one parent, grandparents, relatives or friends [1,2]. According to the data of the Ministry of Civil Affairs of China, in August 2019, there were more than 6.79 million LBC in rural areas in China. In terms of guardianship, 96% of the LBC in rural areas are cared for by their grandparents, and 4% of LBC are cared for by other relatives and friends. As for the age distribution of the LBC, 21.7% are aged 0-5 years old, 67.4% are aged 6–13 years old, and 10.9% are aged 14–16 years old. In terms of regional distribution, Sichuan province has the largest number of LBC, followed by Anhui, Hunan, Jiangxi, Hubei and Guizhou provinces [3]. There were 150,019 LBC in Shandong by the end of 2016, accounting for 1.66% of the total number in China (9.02 million).

LBC is a phenomenon not only in China, but also around the world. However, due to China’s deep-rooted urban–rural dual structure, the phenomenon of LBC is particularly prominent in China. The large amount of LBC in China has attracted more and more attention from China’s educators, sociologists and criminologists, and even government and education departments. More and more scholars come to study on the mental health of LBC, parent-child relationship, the relationship between social support and mental health, mental health and subjective well-being, mental health and left-behind experience, education and health management, and so on.

Elder’s life course theory shows that the changes and events of the family in the life course of the individual from childhood to adulthood, marriage, parenthood to old age will have an impact on the individual development. In the book *Children of the Great Depression*, Elder also pointed out that the personal life course was embedded in the historical time and the events they experience in their life years, and was also shaped by these time and events. The series of life changes or life events will have an important influence on the individual’s development [4]. LBC, compared with children who do not have left-behind experience, due to the special growth environment during the life process, their character would be affected or changed under the pressure in the special environment. Many studies have reported that, compared with a single-parent family or the absence of both parents, both parents’ company is more conducive to children’s growth and development [5], and parental absence has a detrimental impact on the emotional and behavioral functioning of children [6,7,8,9]. Physical inaccessibility and lack of communication tends to disrupt parent–child attachment, leading most LBC to live in a situation lacking parental care, support, guidance and communication, which may have many negative emotional impacts on the children [10,11,12,13]. Li’s study indicated that if children failed to receive proper guidance and help from their parents in ideology and values during the critical period of their growth and development, the normal development of their personality would be affected. Compared with ordinary children, due to lack of proper guidance and help from their parents during the left-behind time, many of the LBC have no opinions or dare to express themselves. especially when they encounter things or face problems, they would lack judgment [14]. Usually, these LBC had less communication with their parents during the left-behind period, and the content of the communication was monotonous. More attention from the parent was paid to the child’s physical health and academic performance, and less was paid to their mental health issues, which caused different levels of mental health problems, for example, depression, sensitive interpersonal relationships, social anxiety, etc. [15]. A meta-analysis of social anxiety in left-behind children in rural areas of China shows that rural LBC’s social anxiety was 36.1% (N = 18544), which was higher than that of non-left-behind children (20.2%) [16], and LBC had a lower social adaptation than non-left-behind children [17].

Some of these LBC study hard and have a chance to be college students, with left-behind experience. The definition of college students with left-behind experience was first proposed by Zhang, who defined this group as college students who had left-behind experience before they went to college [2]. So far, the definition of these left-behind students is not consistent. Wen and Zeng defined them as children: (1) whose parents (one or both) go away from home for a long time, migrating to other cities for work, business or study or (2) who have been separated from their parent(s) for 6 months or more, and (3) whose current age is 17–21 years old [18]. He defined them as children: (1) whose parents (one or both) go out to work during their growth stage, leaving them in their hometown; (2) who had a left-behind experience before they were 17 years old; and (3) who have been separated from their parent(s) for more than 6 months and are now studying at university [19]. Jin defined them as children: (1) whose parents (one or both) go out to work during their growth stage, leaving them in their hometown; (2) who had a left-behind experience before they were 14 years old; and (3) who were separated from their parent(s) for half a year or more each time [20].

In a study on college students with left-behind experience, Zhang first summarized the mental health characteristics of these students in 2006. She then conducted a survey in 2008 through qualitative interviews and psychological consultation, which indicated that left-behind experience had a certain impact on the psychological health of college students. The impacts were different between boys and girls: Girls were more worried about safety issues than boys. Meanwhile, through interviews, Shu and Zhang found that if the girls took on the responsibility of caring for their siblings, they were often brave, and the impact of left-behind experience on safety and horror was positive. For example, some left-behind students were more independent and more considerate of other people’s feelings and showed higher inverse quotient indices [2,21,22,23,24]. Besides lacking parental companions and having low family conditions and growing environments, as well as unsound social support networks, these left-behind students tended to be more sensitive than the students who did not have left-behind experience [25,26,27]. Some studies found that left-behind experience had significant effects on college students’ psychological development and mental health in adulthood. Left-behind students were more likely to develop mental health problems, such as psychological imbalance and emotional disorders, a fragile mentality, and affective disturbance [24,28,29]. They seemed to be more sensitive in interpersonal communication and to feel more inferior. Some students had lower self-esteem, more negative emotions, fewer positive coping styles, low life satisfaction, low subjective well-being and so on [30].

In order to close some of these research gaps related to the mental health of college students with left-behind experience, we performed a cross-sectional comparative study of college students with left-behind experience and controls (college students without left-behind experience), in which we assessed the prevalence rate of mental health, including total mental health, somatization, obsessive-compulsion (OCD), international sensitive, depression, anxiety, hostility, terror, paranoia, psychosis and other symptoms. We examined demographic and social support factors, including gender, rural or urban source, only child or not, separation type with their parents, duration of being left-behind, education level of the parents and the people they lived with during the left-behind time, their relationship with people, and so on. Therefore, based on the life course theory and the special growth experience of the college students, we established the following three hypotheses.

**Hypothesis** **1.**
*Whether the left-behind experience is a significant risk factor for the mental health of college students.*


**Hypothesis** **2.**
*Mental health problems are more prevalent in college students with left-behind experiences than controls, and the prevalence of mental health problems among left-behind students is higher than that of controls, including the prevalence of the dimensions of the SCL-90 scale.*


**Hypothesis** **3.**
*Different left-behind experience characteristics and social supports have different effects on the mental health of college students, not only on the total mental health, but also on the ten dimensions of the SCL-90 scale.*


## 2. Material and Methods

### 2.1. Paticipants

Based on the definition of LBC and China’s College Entrance Examination Policy (allowing social groups to participate in the college entrance examination), we finally adopted Zhang’s definition of college students with left-behind experience: as long as you have had a left-behind experience before going to college, you will be defined as a college student with left-behind experience [2].

A cross-sectional comparative design was conducted in Yantai of Shandong Province, China. Yantai is a coastal city with a population of about 7 million, and the economic level is in the third place in Shandong province and the 20th in China. Due to Yantai’s better economy, geography and education, larger numbers of college students are attracted from other provinces, especially some central and western provinces, where there are a large number of left-behind children. There are five undergraduate universities in Yantai. Considering the distribution of the subject branch and representativeness of the sample, we selected three different types of universities (a comprehensive university, a normal university and a medical university) and nineteen undergraduate majors that were recruited nationwide.

In order to calculate the sample size by considering the prevalence rate of the mental health problems among college students with left-behind experience, we conducted a pre-survey in one medical university, with a sample size of 455 in Yantai city [30]. The prevalence of mental health problems among the left-behind students was 54.7% in the pre-survey. We chose a 95% confidence level (*α*), allowable error (*δ*) of 3% and a computational formula $N=\frac{Z_{\alpha /2}^{2}\times P\times \left(1-P\right)}{\delta^{2}}$. When *P* = 0.5 and *δ* = 3%, *N* was the largest sample size, with 1057. The proportion of left-behind students with left-behind experience in the pre-survey was 30.1%, which was higher than that in some similar investigations (nine studies, n = 20,565) [31,32]. In order to ensure the sample size and representativeness of the left-behind students, we thus expanded the final target sample size by 30% (N = 1374). To account for invalid questionnaires and other potential limitations, based on the percentage of valid questionnaires of 91% in the pre-survey, we expanded the final target sample size by 30% and recruited 1650 college students. Finally, we received 1605 valid questionnaires, with an effective rate of 97.27%. Among these 1605 participants, there were 312 left-behind college students, with a proportion of 19.44%, which was much lower than that in the pre-survey.

### 2.2. Procedure

This study was approved by the Ethics Committee of Binzhou Medical University (Human Research Ethics Committee (No. 2019-49), founded approved this study in 2019), with the help and approval from three university officials and tutors. The questionnaires was filled out anonymously, and the consent of the respondents was gained before the investigation. The questionnaires were assigned to each student and completed in a classroom administrated by the investigators and assistants, who were trained professionally before the investigation. Under the consent of the participants, we investigated each class as one unit in order to eliminate the worries of the participants. The students were explicitly assured that their responses would be treated anonymously and that the research data would be stored securely. In the process of completing the questionnaires, the investigators and assistants walked around the classroom to remind the participants not to communicate or help each other. If the students had difficulties, they could ask the investigators or assistants for help, not their classmates. The questionnaires were finished and checked by the investigators or assistants, before being submitted.

### 2.3. Measurement

We chose the Symptom Checklist 90 (SCL-90), compiled by the famous American psychologistDerogatis in 1977, as the investigating tool [33,34]. As one of the most well-known checklists of psychological symptoms, SCL-90 has a high validity and reliability for measuring mental health, and it contains a list of symptoms that are divided into different psychological conditions, such as somatization, obsessive-compulsion (OCD), interpersonal sensitivity, depression, anxiety, hostility, terror, paranoia, psychosis and other symptoms. However, a number of studies have shown that SCL-90 and its variants tend to measure general psychology and mental health problems. It is suitable for participants over 16 years of age. The items of SCL-90 are rated on a five-point Likert scale, and higher scores indicate more mental health problems. According to the Chinese norm, if the total score of the SCL-90 scale is more than 160 points, this indicates a subject with positive mental health problems, as do positive responses to any dimension/factor’s mean score of more than 2 points (the total score of each dimension/factor, divided by its item number). In this study, the Cronbach *α* is 0.982, which indicates it has a good reliability.

### 2.4. Statistical Analysis

First, a chi-squared test was used to assess the differences between left-behind students and controls in terms of mental health prevalence and equilibrium test demographic characteristics. Then, the coarsened exact matching (CEM) algorithm was used for data matching in order to control the imbalance between the two groups (left-behind students and controls). CEM is not a method of estimation, but rather a way to preprocess a dataset so that the estimation based on the matched data will be less ‘model-dependent’ (i.e., less a function of apparently small and indefensible modelling decisions) than when based on the original full dataset [35,36,37]. Matching involves pruning observations that have no close matches on covariates in both the treated and control groups; it can improve the estimation of causal effects in observational studies by reducing imbalance in covariates between the treated and control group [35,38,39].

The CEM algorithm involves three steps. Firstly, each covariate variable (X) is coarsened by recoding to the group and appointing the indistinguishable values with the same value [40]. The second step: using the algorithm of exact matching to data matching, and removing the coarsened data, and make sure that the last remaining data should be matched data [37,38,39,40,41,42]. The third step: calculate a comprehensive multivariate imbalance measure. The measure is based on the *L*1 difference between the multivariate histograms of all the pretreatment covariates in the treated group and that in the control group [39,41]. *L*1 provides an easy interpretation: Conditioned on the coarsening level, if the empirical distributions before and after CEM are completely separated, then *L*1 = 1, while if the distributions perfectly overlap, then *L*1 = 0; otherwise *L*1 ranges from 0 (perfect global balance) to 1 (maximal imbalance). A good matching substantially reduces *L*1 [40,41]. *L*1 can be calculated as follows. Formula (a): Firstly, covariates were coarsened into bins. Then, the discretized variables were cross-tabulated as *X*_1_ ×…… × *X_k_* for the treated and the control groups separately, and the *k*-dimensional relative frequencies were recorded for the treated and *fε*1*……εk* for the control *gε1……εk* units. Finally, the measure of imbalance is the absolute difference over all the cell values [39,40,41]:(1)$$L_{1(f,g)}=\frac{1}{2}\sum {}_{\varepsilon 1\ldots \ldots \varepsilon k}\left|f_{\varepsilon 1\ldots \ldots \varepsilon k}\right|$$

We modelled the CEM using the *cem* command code with Stata 13.0 analysis software. More details of the CEM method can be found in other literature.

## 3. Results

### 3.1. Demographic Characteristics of the Participants

In this study, 1605 college students from three universities were investigated: 312 college students with left-behind experience, and 1293 non-left-behind students were used as controls. The demographic characteristics of left-behind students and controls are shown in Table 1. Equilibrium tests of the demographic characteristics of left-behind students and controls showed significant differences in terms of the university (*Χ*^2^ = 15.74, *p* < 0.001), gender (*Χ*^2^ = 13.56, *p* < 0.001), and urban or rural source (*Χ*^2^ = 27.33, *p* < 0.001), which indicated that the variables for the two groups were unbalanced.

### 3.2. Data Matching Performance

Equilibrium tests showed that the multivariate *L*_1_ statistics was 0.306 of the 1605 observations in the database (312 left-behind students and 1293 controls). Then, we used university, gender, grade, only child or not and urban or rural sources as the control variables for data matching; after matching, 100 observations were deleted, and 1505 matched observations remained, including 311 left-behind students and 1194 controls. The multivariate *L*_1_ between left-behind students and controls was 4.84 × 10^−15^, actually close to zero, which was much lower than that before matching (Table 2). Table 3 showed there were no statistical differences in terms of the control variables between the two groups, which indicated good matching performances and thus the two groups became more comparable.

### 3.3. Comparison of the Prevalence of Mental Health Problems in Left-Behind Students and Controls

Controlling the confounding factors, we used matched data to compare the mental health problems of the two groups. The prevalence rate of total mental health problems was 35.69% among left-behind students, which is substantially higher than the prevalence of 19.68% among controls (*OR* = 2.27, 95%*CI*: 1.73 to 2.97). Similarly, left-behind students showed a higher prevalence rate of the SCL-90 scale’s ten dimensions: somatization problems, 20.58% vs. 9.38% (*OR* = 2.50, 95%*CI*:1.79 to 3.51); Ocd, 44.05% vs. 28.56% (*OR* = 1.97, 95%*CI*:1.52 to 2.55); interpersonal sensitivity, 40.51% vs. 22.19% (*OR* = 2.39, 95%*CI*:1.83 to 3.11); depression, 29.90% vs. 17.33% (*OR* = 2.03, 95%*CI*:1.53 to 2.71); and other dimensions, which are shown in Table 4.

### 3.4. Risk Factors Analysis for the Mental Health of College Students with Left-Behind Experience

Table 5 shows the risk factors for the total mental health and ten dimensions of the SCL-90 scale, mainly including demographic characteristics, different left-behind experiences and social support variables. (1)Regarding the left-behind length of time variable: less than two years relative to more than two years was *OR* = 0.47 and 95%*CI*: 0.26 to 0.87; Regarding the variable of ‘the reasons your parents are not around’:

Father absence had an important influence on college students’ mental health, and father’s absence relative to ‘other reasons’ in total mental health problems was *OR* = 1.97 and 95%CI: 1.08 to 3.60, OCD was *OR* = 2.17 and 95%CI: 1.24 to 3.81,anxiety was *OR* = 2.67 and 95%*CI*: 1.34 to 5.33, terror was *OR* = 2.03 and 95%*CI*: 1.19 to 3.47, and paranoia was *OR* = 1.94 and 95%*CI*: 1.12 to 3.35. (2)The variable of ‘who do you live with during the left-behind time’ was an important risk factor for total mental health, somatization, international sensitive, depression and paranoia. Compared with living alone, or living with other people, the students living with father or mother during the left-behind time were less likely to have mental health problems (Total mental health: *OR* = 0.42, 95%*CI*: 0.22 to 0.80; somatization (*OR* = 0.46, 95%*CI*: 0.22 to 0.98); international sensitive (*OR* = 0.56, 95%*CI*: 0.34 to 0.91); depression (*OR* = 0.51, 95%*CI*: 0.31 to 0.84); anxiety (*OR* = 0.52, 95%*CI*: 0.31 to 0.88; paranoia (*OR* = 0.52, 95%*CI*: 0.31 to 0.88); psychosis (*OR* = 0.59, 95%*CI*: 0.35 to 0.98). (3)The education level of the people the students live with during the left-behind time was an significant risk factor on the total mental health (*OR* = 0.74, 95%*CI*: 0.56 to0.98), OCD(*OR* = 0.71, 95%*CI*: 0.54 to 0.93) and paranoia (*OR* = 0.74, 95%*CI*: 0.57 to 0.95). The higher the education level of the people the students live with during the left-behind time, the lower the prevalence rate of the total mental health problems, OCD and paranoia problems. The influences of other variables were all shown in Table 5.

## 4. Discussion

This study makes an important contribution to the domestic and international literature on the mental health of college students with left-behind experience. The results of this study confirmed that left-behind experience was a significant risk factor influencing college students’ mental health. The mental health problems were more prevalent in left-behind students than in controls, which confirmed the first hypothesis of our study, and the prevalence rate was higher than that of the norm for Chinese college students [42,43]. These findings were consistent with the studies of Zhang and Wen, which provided evidence that left-behind students had limited social support networks that endangered their mental health problems [2,18]. Some studies indicated that due to a lack of family guidance and support during childhood, these left-behind college students did not develop the ability to manage their negative emotions well, which affected their mental health and caused some psychological problems for a long time [44,45]. Sigmund Freud, the originator of psychoanalysis, also pointed out that if individuals’ emotional needs are not met during childhood, negative emotions accumulate and are buried in the subconscious, which affects these individuals’ mental health [45,46]. Notably, however, Li found that college students with left-behind experience behaved in more modest and cautious ways to correspond with the requirements of traditional Chinese cultures [24].

In this study, we found that the university, gender, only child or not and urban or rural source variables were confounding factors in the comparison of the mental health of left-behind students and controls, and left-behind students had clear mental health and psychological problems, which were reflected in the levels for the ten dimensions of the SCL-90 scale (somatization, OCD, interpersonal sensitivity, hostility, terror, paranoia, etc.), which confirmed the second hypothesis of our study. The result also indicates that these confounding factors are the influencing factors of left-behind students’ mental health, which is consistent with previous studies [22,23,47,48]. Besides, the prevalence of OCD among college students with left-behind experience was 44.05%, but it was 28.56% among controls. The depression rate of 29.90% in college students with left-behind experience is higher than the 17.33% among controls in our study, and it is also higher than 15.3% [49], 24.8% [50] and 14.1% [51], which are the rates for LBC. These college students always show significantly less energy, life satisfaction and emotional and behavioral control than controls, which affects their mental health [12,47,48,49].

Different left-behind experiences and kinds of social support have different effects on college students’ mental health (Table 5), which confirmed our third hypothesis. Because of the nature of professions, the burden of learning and the learning atmosphere, left-behind students in medical universities were more prone to anxiety, paranoia and psychosis problems, while left-behind students in comprehensive universities were more prone to hostility, paranoia and psychosis problems. Besides, the impact of left-behind experience was greater among boys than among girls, and left-behind experience had more positive effects on girls’ mental health, which was consistent with some previous studies [52,53,54]. Since girls mature earlier than boys, they are more likely to adopt the role of parent and be more independent in processing and responding to events, especially negative events. Our study found that college students living alone during their left-behind time were more prone to mental health problems than those who lived with one of their parents. Students lacking of fathers’ company were especially more prone to OCD, anxiety, terror and paranoia problems, which reiterated the importance of parental companionship for children and adolescents’ mental health [17,55,56]. Living with a father or a mother, students will receive more care and love, and in this situation, they can receive greater education on how to communicate with others. Especially when their fathers are around, they will be braver, more secure, and have a certain amount of emotional guidance and release. Thus, father or mother’s companionship and a warm mother–child relationship may, to a certain degree, satisfy the children by giving them the strength to resist lonely experiences, which will have a potential impact on their mental health [28,55,57,58,59,60].

If the children’s parents migrated or the children lived with a support family or single-parent family, and if they lacked an intimate parent–child relationship and communication, they were not able to be trained to have the ability to communicate with others [61,62]. As dynamic and emotion-related traits spanning from early childhood, psychoticism and neuroticism are partially affected by life experiences and the external environment [63]. Thus, the stressful experience of being left-behind would lead to a susceptibility to negative emotions, such as anxiety, fear of contacting others, having few friends, feeling loneliness, and not venting feelings [64,65]. Then the left-behind students could not trust others and establish a close relationship with people around them. When they built close relationships with the people around, they had a lot of worries and troubles. As a result, college students with left-behind experience are prone to anxiety, depression, compulsion, interpersonal disorder, a lower subjective well-being, and an incapacity to communicate well with other people and establish meaningful relationship [27,66]. The results of this study also showed that having a good relationship with their teachers and classmates was a protective factor for somatization, OCD, depression and hostility problems.

## 5. Limitations

The results of this study provide some data and basis for the study of left-behind college students’ mental health. To make the conclusion more reliable, we applied coarsened exact data matching to compare these college students with controls. However, our study also had some limitations. First, due to being a cross-sectional study, the evidence of this study was weak in verifying causality, and there also could be some unobserved confounding factors in this study. Cohort studies are therefore needed in the future research. Second, despite our efforts to ensure the representativeness of the sample and the investigative quality of the study, we selected participants from only one city in Eastern China, rather than nationwide, which reduced our ability to generalize the findings to college students with left-behind experience in other areas of China. Future studies with larger and national samples should be conducted to make the samples more representative.

## 6. Conclusions

Despite these limitations, this study advances our understanding of the mental health of college students with left-behind experience and provides data to guide the development of interventions and prevention programs for promoting the mental health of college students. By controlling the confounding factors using CEM, our results indicated that left-behind experience was an important influencing factor for mental health problems among college students. Because of the limited social support during their left-behind time, these left-behind students showed a higher prevalence than controls not only in total mental health problems, but also in somatization, obsessive-compulsion (OCD), depression, anxiety, hostility, terror, paranoia and psychosis problems.

## Figures and Tables

**Table 1 ijerph-17-01511-t001:** Equilibrium test of the demographic characteristics of the two groups.

Variables	Options	Left-Behind Students (N, %)	Control (N, %)	χ^2^-Value	*p*-Value
University	Medical university	131 (42.0)	639 (49.4)	15.47	<0.001
	General university	76 (24.2)	354 (27.4)		
	Comprehensive university	105 (33.7)	300 (23.2)		
Gender	male	124 (39.7)	375 (29.0)	13.56	<0.001
	female	188 (60.3)	918 (71.0)		
Grade	freshman	108 (34.6)	467 (36.1)	0.076	
	sophomore	83 (26.6)	360 (27.8)		
	junior	92 (29.5)	397 (30.7)		
	senior	29 (9.3)	69 (5.3)		
Urban or rural source	urban	63 (20.2)	461 (35.7)	27.33	<0.001
	rural	249 (79.8)	832 (64.3)		
Only child or not	only child	110 (35.3)	519 (40.1)	2.52	0.113
	not only child	202 (64.7)	774 (59.9)		

Note: N (%) was reported, and a chi-square test was used as an equilibrium test. All statistical tests were two-tailed, and the threshold of significance was defined as *p* < 0.05.

**Table 2 ijerph-17-01511-t002:** The *L*_1_ measure of imbalance before and after Coarsened Exact Matching.

Variable	Before Matching: *L*_1_ (mean)	After Matching: *L*_1_ (mean)
University	0.108 (0.182)	3.6 × 10^−15^ (−4.2 × 10^−15^)
Gender	0.107 (−0.107)	3.0 × 10^−15^(−8.0 × 10^−15^)
Grade	0.040 (0.082)	4.9 × 10^−15^(−1.5 × 10^−14^)
Only child or not	0.049 (0.049)	2.8 × 10^−15^(−5.6 × 10^−15^)
Urban or rural source	0.155 (−0.155)	2.0 × 10^−15^(−4.0 × 10^−15^)
Multivariate *L*_1_	0.306	4.84 × 10^−15^

Note: *L*_1_ was computed for measuring the imbalance between left-behind students and controls, using Stata 13.0.

**Table 3 ijerph-17-01511-t003:** Equilibrium test of demographic characteristics between the two groups after data matching.

Variables	Options	Left-Behind Students (N, %)	Controls (N,%)	χ^2^-Value	*p*-Value
University	medical university	131 (42.12)	503 (42.12)	<0.001	1.000
	general university	75 (24.12)	288 (24.12)		
	comprehensive university	105 (33.76)	403 (33.76)		
Gender	male	123 (39.55)	472 (39.55)	<0.001	0.995
	female	188 (60.45)	722 (60.45)		
Grade	freshman	108 (34.73)	415 (34.73)	<0.001	1.000
	sophomore	82 (26.37)	315 (26.37)		
	junior	92 (29.58)	353 (29.58)		
	senior	29 (9.32)	111 (9.32)		
Urban or rural source	urban	62 (19.94)	238 (19.94)	<0.001	0.999
	rural	249 (80.06)	956 (80.06)		
Only child or not	only child	110 (35.37)	422 (35.37)	<0.001	0.993
	not only child	201 (64.63)	772 (64.63)		

Note: N (%) is the number of students in each group and the percentage. A chi-square test(χ^2^-value) was used as an equilibrium test of the two groups. All statistical tests were two-tailed, and the threshold of significance was defined as *p* < 0.05.

**Table 4 ijerph-17-01511-t004:** Comparison of the prevalence of mental health problems in left-behind students and controls after data matching.

Dimension of SCL-90 Scale	Left-Behind Students (n = 311)	Controls (n = 1194)	*OR* (95% *CI)*	*p*-Value
	Positive Cases (N, %)	Positive Cases (N, %)		
Somatization	64 (20.58)	112 (9.38)	2.50 (1.79,3.51)	<0.001
OCD	137 (44.05)	341 (28.56)	1.97 (1.52,2.55)	<0.001
Interpersonal sensitivity	126 (40.51)	265 (22.19)	2.39 (1.83,3.11)	<0.001
Depression	93 (29.90)	207 (17.33)	2.03 (1.53,2.71)	<0.001
Anxiety	83 (26.69)	160 (13.40)	2.35 (1.74,3.18)	<0.001
Hostility	97 (31.19)	204 (17.09)	2.20 (1.66,2.92)	<0.001
Terror	85 (27.33)	163 (13.65)	2.38 (1.76,3.21)	<0.001
Paranoia	90 (29.94)	169 (14.15)	2.47 (1.84,3.32)	<0.001
Psychoticism	78 (25.08)	151 (12.65)	2.31 (1.70,3.15)	<0.001
Other symptoms	86 (27.65)	175 (14.66)	2.23 (1.66,2.99)	<0.001
Total mental health	111 (35.69)	235 (19.68)	2.27 (1.73,2.97)	<0.001

Note: After data matching, there were 311 left-behind students and 1194 controls. Positive cases of 10 dimensions: mean score of each dimension is more than 2 points (the total score of a dimension, divided by the item number of this dimension). Positive cases of total mental health: total score is more than 160 points, indicating positive mental health problems. A chi-square test was used to compare the differences between the two groups (two-tailed). The threshold of significance was defined as *p* < 0.05.

**Table 5 ijerph-17-01511-t005:** Multinomial logistics regression to identify mental health risk posed by left-behind experiences among college students with left-behind experience (N = 312).

Variables	Total Mental Health		Somatization		Ocd		Interpersonal Sensitivity		Depression		Anxiety		Hostility		Terror		Paranoia		Psychosis	
	*OR*	95%*CI*	*OR*	95%*CI*	*OR*	95%*CI*	*OR*	95%*CI*	*OR*	95%*CI*	*OR*	95%*CI*	*OR*	95%*CI*	*OR*	95%*CI*	*OR*	95%*CI*	*OR*	95%*CI*
**university** Medical university	1	-	1	-	1	-	1	-	1	-	1	-	1	-	1	-	1	-	1	-
General university	0.87	(0.53, 1.43)	0.65	(0.34, 1.25)	0.70	(0.46, 1.07)	0.87	(0.59, 1.31)	0.78	(0.52, 1.16)	**0.58 ***	**(0.38, 0.89)**	0.77	(0.51, 1.15)	0.69	(0.45, 1.06)	**0.53 ****	**(0.35, 0.81)**	**0.61 ***	**(0.40, 0.94)**
Comprehensive university	1.17	(0.76, 1.81)	1.22	(0.73, 2.04)	1.39	(0.92, 2.09)	1.40	(0.95, 2.05)	1.21	(0.84, 1.75)	1.45	(0.98, 2.15)	**1.46 ***	**(1.01, 2.13)**	1.20	(0.82, 1.75)	**1.84 ****	**(1.24, 2.74)**	**1.61 ***	**(1.10, 2.37)**
**gender** (male:female)	**0.50 ***	**(0.25, 0.96)**	**0.41 ***	**(0.20, 0.87)**	1.11	(0.58, 2.14)	1.04	(0.57, 1.92)	0.81	(0.45, 1.45)	0.64	(0.34, 1.17)	0.94	(0.52, 1.72)	0.81	(0.45, 1.46)	1.36	(0.73, 2.52)	1.06	(0.58, 1.94)
**rural or urban** (rural:urban)	1.11	(0.49, 2.50)	1.56	(0.63, 3.87)	0.97	(0.45, 2.08)	1.26	(0.62, 2.55)	1.21	(0.60, 2.42)	**2.28 ***	**(1.09, 4.78)**	1.86	(0.91, 3.81)	1.30	(0.64, 2.63)	0.74	(0.18, 3.04)	1.08	(0.53, 1.94)
**only child or not** (yes:no)	1.28	(0.65, 2.54)	1.27	(0.60, 2.70)	1.06	(0.55, 2.03)	0.96	(0.53, 1.77)	1.48	(0.81, 2.71)	1.54	(0.81, 2.92)	1.07	(0.58, 1.97)	1.00	(0.54, 1.84)	1.03	(0.57, 1.84)	1.35	(0.73, 2.49)
**parents divorced** (yes:no)	0.38	(0.05, 2.66)	0.17	(0.01, 2.67)	0.31	(0.08, 1.13)	0.50	(0.13, 1.94)	0.23	(0.04, 1.28)	1.47	(0.37, 5.90)	0.69	(0.17, 2.77)	0.81	(0.19, 3.48)	0.74	(0.18, 3.04)	0.55	(0.13, 2.27)
**left-behind time** (<2 years:≥2 years)	0.59	(0.30, 1.17)	0.62	(0.28, 1.37)	1.43	(0.77, 2.68)	0.85	(0.48, 1.51)	0.70	(0.40, 1.22)	**0.47 ***	**(0.26, 0.87)**	0.87	(0.49, 1.54)	0.68	(0.38, 1.21)	1.03	(0.57, 1.84)	0.62	(0.35, 1.14)
**who were absence during the left-behind time?** Other reasons	1	-	1	-	1	-	1	-	1	-	1	-	1	-	1	-	1	-	1	-
Mother	0.90	(0.29, 2.78)	1.25	(0.39, 4.01)	0.28	(0.10, 0.77)	0.73	(0.29, 1.81)	0.87	(0.35, 2.14)	0.23	(0.05, 1.42)	0.64	(0.24, 1.68)	0.69	(0.25, 1.89)	0.53	(0.19, 1.45)	0.73	(0.27, 1.97)
Father	**1.97**	**(1.08, 3.60)**	1.26	(0.65, 2.44)	**2.17 ****	**(1.24, 3.81)**	1.29	(0.78, 2.15)	1.55	(0.93, 2.58)	**2.67 ****	**(1.34, 5.33)**	1.34	(0.80, 2.26)	**2.03 ****	**(1.19, 3.47)**	**1.94 ***	**(1.12, 3.35)**	1.40	(0.83, 2.36)
Both parents	0.77	(0.42, 1.42)	0.82	(0.43, 1.60)	1.29	(0.75, 2.22)	0.93	(0.56, 1.55)	0.90	(0.54, 1.48)	1.21	(0.61, 2.40)	0.98	(0.59, 1.64)	0.84	(0.49, 1.44)	0.98	(0.52, 1.85)	0.75	(0.44, 1.27)
**Who do you live with during left-behind time?** Live alone	1	-	1	-	1	-	1	-	1	-	1	-	1	-	1	-	1	-	1	-
Relatives	1.09	(0.54, 2.19)	0.85	(0.38, 1.89)	1.86	(0.87, 3.99)	1.64	(0.85, 3.19)	1.54	(0.83, 2.87)	1.08	(0.57.2.07)	1.07	(0.57, 2.01)	0.84	(0.45, 1.58)	0.98	(0.52, 1.85)	1.30	(0.70, 2.40)
Grandparents	1.18	(0.73, 1.88)	1.17	(0.68, 2.02)	0.70	(0.44, 1.13)	1.06	(069, 1.64)	1.05	(0.69, 1.59)	1.23	(0.80, 1.91)	1.22	(0.80, 1.86)	1.27	(0.84, 1.93)	**1.61 ***	**(1.04, 2.50)**	1.08	(0.71, 1.65)
Father or mother	**0.42 ****	**(0.22, 0.80)**	**0.46 ***	**(0.22, 0.98)**	0.61	(0.36, 1.04)	**0.56 ***	**(0.34, 0.91)**	**0.51 ****	**(0.31, 0.84)**	**0.52 ***	**(0.31, 0.88)**	0.71	(0.44, 1.17)	0.62	(0.36, 1.04)	**0.52 ***	**(0.31, 0.88)**	**0.59 ***	**(0.35, 0.98)**
**Education level of the parents** Illiteracy Junior middle secondary school Senior middle school College and above	0.79	(0.55, 1.13)	0.75	(0.50, 1.12)	0.96	(0.67, 1.37)	0.90	(0.65, 1.24)	0.91	(0.66, 1.25)	0.87	(0.62, 1.21)	0.85	(0.62, 1.18)	0.83	(0.61, 1.15)	0.77	(0.55, 1.08)	0.81	(0.59, 1.12)
**Education level of the people you live with** Illiteracy Primary school Junior middle l secondary school Senior middle school College and above	**0.74 ***	**(0.56, 0.98)**	0.79	(0.58, 1.09)	**0.71 ***	**(0.54, 0.93)**	0.83	(0.65, 1.06)	0.87	(0.68, 1.10)	0.85	(0.66, 1.09)	0.91	(0.71, 1.15)	0.84	(0.66, 1.07)	**0.74**	**(0.57, 0.95)**	0.83	(0.65, 1.07)
**Do you think the people around care about you?** Care very much	1	-	1	-	1	-	1	-	1	-	1	-	1	-	1	-	1	-	1	-
Care occasionally	1.26	(0.71, 2.26)	1.05	(0.58, 2.10)	1.80	(0.92, 3.49)	1.70	(0.93, 3.09)	0.93	(0.53, 1.61)	0.98	(0.55, 1.75)	1.54	(0.86, 2.75)	**1.72 ***	**(1.01, 2.97)**	1.68	(0.94, 3.01)	1.54	(0.89, 2.69)
Little care	1.68	(0.79, 3.60)	2.03	(0.89, 4.62)	1.10	(0.52, 2.34)	1.13	(0.56, 2.27)	1.37	(0.68, 2.77)	1.41	(0.68, 2.96)	1.11	(0.54, 2.24)	0.68	(0.36, 1.27)	0.75	(0.36, 1.55)	1.20	(0.61, 2.39)
Never care	0.77	(0.37, 1.63)	0.83	(0.37, 1.85)	0.63	(0.32, 1.25)	0.83	(0.43, 1.59)	1.24	(0.64, 2.39)	1.15	(0.58, 2.32)	1.04	(0.54, 2.04)	0.57	(0.28, 1.13)	1.17	(0.57, 2.37	1.07	(0.56, 2.06)
**The relationship with the teachers and classmates** Very poor	1	-	1	-	1	-	1	-	1	-	1	-	1	-	1	-	1	-	1	-
Not good	2.20	(0.69, 7.00	2.21	(0.65, 7.51)	3.48	(0.59, 20.38)	2.17	(0.58, 8.03)	1.44	(0.43, 4.82)	0.71	(0.12, 4.24)	4.91	(0.88, 27.49)	2.33	(0.71, 7.66)	2.71	(0.67, 10.89)	3.51	(0.91, 13.55)
average	**0.47 ***	**(0.24, 0.92)**	**0.32 ****	**(0.15, 0.70)**	0.47	(0.20, 1.13)	0.97	(0.51, 1.86)	0.70	(0.35, 1.39)	0.63	(0.26, 2.04)	0.61	(0.28, 1.32)	0.68	(0.36, 1.27)	0.60	(0.30, 1.21)	0.75	(0.39, 1.45)
good	**0.36 ****	**(0.17, 0.78)**	**0.35 ***	**(0.14, 0.83)**	**0.38 ***	**(0.15, 0.96)**	0.78	(0.38, 1.57)	**0.45 ***	**(0.21, 0.96)**	1.23	(0.69, 2.18)	**0.40 ***	**(0.18, 0.90)**	0.56	(0.28, 1.13)	0.63	(0.29, 1.34)	0.60	(0.29, 1.22)
**Whether ave good friends or not in school?** (yes:no)	0.69	(0.33, 1.44)	0.98	(0.42, 2.28)	0.68	(0.31, 1.48)	0.52	(0.25, 1.06)	0.92	(0.46, 1.81)	0.80	(0.39, 1.64)	0.96	(0.48, 1.91)	0.68	(0.35, 1.33)	0.54	(0.26, 1.12)	0.67	(0.34, 1.34)

Note: (1) Influencing factors mainly including the variables of demographic characteristic and social support of the left-behind students during the left-behind time. Mental health mainly including the total mental health and the ten dimensions of SCL-90 scale (The 10th dimension was excluded, because all the variables had no effect on it). (2) All statistical tests were two-tailed. * Significance at the 0.05 level (two-tailed); ** Significance at the 0.01 level (two-tailed). Adjusted for all other variables in the analysis. *OR* (odd ratio) and 95%*CI* (95% confidence interval) were calculated using binary logistics regression. The first group of each independent variable is the control group. The variables of the education level of students’ parents and the people they live with are ordered variables.

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
