# Peer review of "The Influence of Left-Behind Experience on College Students’ Mental Health: A Cross-Sectional Comparative Study"

_ijerph, 2020, doi:10.3390/ijerph17051511_

Round 1

Reviewer 1 Report

There has been a great interest in the group of left-behind children in social scientific research. However, very few research has examined its mental health consequences especially for their later life (i.e., college students). In this regard, this study make a great contribution to the literature.

There are a couple of minor points I would like to comment on.

Abstract: remove repetitive words of OR. I think the author accidently made a typo.

Words of choice: rather than important influencing factor, what about changing it into “ significant risk factor” which may more clearly highlight major findings.

Introduction: page 2 Line 49-50 The instances from other countries were not quite fit into the current context. If the author wanted to place them here they should explain in more details about the similar patterns.

Page 2 Line 54-55 Specifically which aspects of LBC in China have emerged the main issue studied in policy and academic setting.

One major concern about this manuscript is the lack of theoretical background. I think this research can be framed at the intersection of life course and stress process theory just an example.

Measurement: what is the reliability of SCL-90 for the study sample in the current study?

Author Response

Dear reviewer,

    It’s my great honor to receive your comments concerning my article. Thank you for your valuable suggestions. According to your suggestions, I have made modifications in the paper and a point-by-point response to the commonts(Please see the attachment).

Thank you and best regards.

Yours sincerely!

Author: Haixia Liu

Reviewer 2 Report

The strengths of this paper include:

  • A carefully researched literature review, highlighting the issue of LBC.
  • A fairly large sample of LBC in college.
  • Inclusion of a number of demographic and mental health variables, providing a clear picture of the difference between LBC and non-LBC college students.

In terms of areas where the authors could provide additional information or make changes:

  • There are sentences in the literature review that leave the reader with the impression that LBC are damaged or bad. For example, on line 64 you state, “LBC have certain peculiarities, which are due to their left behind-experience.” This is a causal claim that is not supported by the current evidence. Also on line 73 you state, “They would lack judgement and be prone to some bad behaviors.” This appears too broad of a statement given the research we have to date. Many children “lack judgement” because they are children and do not have fully developed brains.
  • Hypothesis 1 and 3 are unclear and possibly worded so that it would difficult to disprove them.
  • The overall results of the study are impressive, with LBC college students faring worse on several mental health measures compared to similarly matched non-LBC college students. However, I found the analyses and presentation of the data to be quite confusing. Table 5 was especially confusing—is there a way to present to most important findings related to your hypothesis and summarize the findings that are less central to your thesis?
  • The syntax could be improved to achieve better flow from section to section and within sections.

Author Response

Dear reviewer,

    It’s my great honor to receive your comments. Thank you for your valuable suggestions. According to your suggestions, I have made modifications in the paper and had my manuscript undergone extensive English editing by MDPI. And I also made point-by-point responses to the comments(Please see the attachment).

Thank you and best regards.

Yours sincerely!

Author: Haixia Liu
